# A phenolic-rich extract from *Ugni molinae* berries reduces abnormal protein aggregation in a cellular model of Huntington's disease

**Rodrigo Pérez-Arancibia**[1,2,3,4], **Jose Luis Ordoñez**[1,5], **Alexis Rivas**[2,3,6], **Philippe Pihán**[2,3,6], **Alfredo Sagredo**[2,3,6], **Ulises Ahumada**[3,4], **Andrés Barriga**[7], **Ivette Seguel**[1], **César Cárdenas**[3,4,8,9], **Rene L. Vidal**[2,3,4], **Claudio Hetz**[2,3,6,9]*, **Carla Delporte**[1]*

**1** Laboratorio de Productos Naturales, Departamento de Química Farmacológica y Toxicológica, Facultad de Ciencias Químicas y Farmacéuticas, Universidad de Chile, Santiago, Chile, **2** Instituto de Neurociencia Biomédica (BNI), Facultad de Medicina, Universidad de Chile, Santiago, Chile, **3** Centro FONDAP de Gerociencia, Salud Mental y Metabolismo (GERO), Santiago, Chile, **4** Centro de Biología Integrativa, Facultad de Ciencias, Universidad Mayor, Santiago, Chile, **5** Laboratorio de Química Inorgánica y Analítica, Departamento de Química Inorgánica y Analítica, Facultad de Ciencias Químicas y Farmacéuticas, Universidad de Chile, Santiago, Chile, **6** Programa de Biología Celular y Molecular, Instituto de Ciencias Biomédicas, Universidad de Chile, Santiago, Chile, **7** Unidad de Espectrometría de Masas, Facultad de Ciencias Químicas y Farmacéuticas, Universidad de Chile, Santiago, Chile, **8** Department of Chemistry and Biochemistry, University of California Santa Barbara, Santa Barbara, CA, United States of America, **9** Buck Institute for Research on Aging, Novato, California, United States of America

* chetz@uchile.cl, chetz@buckinstitute.org (CH); cdelpor@uchile.cl (CD)

**Data Availability Statement:** All relevant data are within the manuscript and its Supporting Information files.

## Abstract

Accumulation of misfolded proteins in the brain is a common hallmark of most age-related neurodegenerative diseases. Previous studies from our group identified the presence of anti-inflammatory and antioxidant compounds in leaves derived from the Chilean berry *Ugni molinae* (murtilla), in addition to show a potent anti-aggregation activity in models of Alzheimer´s disease. However, possible beneficial effects of berry extracts of murtilla was not investigated. Here we evaluated the efficacy of fruit extracts from different genotypes of Chilean-native *U. molinae* on reducing protein aggregation using cellular models of Huntington´s disease and assess the correlation with their chemical composition. Berry extraction was performed by exhaustive maceration with increasing-polarity solvents. An unbiased automatic microscopy platform was used for cytotoxicity and protein aggregation studies in HEK293 cells using polyglutamine-EGFP fusion proteins, followed by secondary validation using biochemical assays. Phenolic-rich extracts from murtilla berries of the 19–1 genotype (ETE 19–1) significantly reduced polyglutamine peptide aggregation levels, correlating with the modulation in the expression levels of autophagy-related proteins. Using LC-MS and molecular network analysis we correlated the presence of flavonoids, phenolic acids, and ellagitannins with the protective effects of ETE 19–1 effects on protein aggregation. Overall, our results indicate the presence of bioactive components in ethanolic extracts from *U. molinae* berries that reduce the load of protein aggregates in living cells.

**Funding:** This work was financially supported by ANID-Chile (Agencia Nacional de Investigación y Desarrollo - www.ANID.cl) grants for Doctoral studies N° 21150769 (RPA) and N°21150851 (JLO), FONDECYT 1130155 (CDV), FUNDACION COPEC-UC grant 2013.R.40 (AR), FONDEQUIP EQM120164 (CH), and Millennium Institute P09-015-F (RLV/CH). We also thank the support from ANID/FONDAP program 15150012 (RLV/CH), FONDECYT 1191003 (RLV), FONDECYT 1180186 (CH), FONDECYT 3190738 (AS), FONDECYT 3210294 (PP), FONDECYT 1200255 (CC), FONDECYT 1140549, FONDEF ID16I10223, FONDEF ID11E1007, CONICYT-Brazil 441921/2016-7, Michael J Fox Foundation for Parkinson´s Research – Target Validation grant 9277 (www.michaeljfox.org/) (CH), European Commission R&D MSCA-RISE 734749 (ec.europa.eu) (CH). In addition, we would like to thank the support from the U.S. Air Force Office of Scientific Research FA9550-16-1-0384, US Office of Naval Research-Global (ONR-G) N62909-16-1-2003 (CH).

**Competing interests:** The authors are co-inventors on a patent application (CH, RP, AR, RLV, CD): Title: Use of leaf extracts from U. Molinae for the treatment of protein misfolding disorders. Chilean application number (INAPI): 3359-201. This does not alter our adherence to PLOS ONE policies on sharing data and materials.

## Introduction

The accumulation of misfolded proteins is the hallmark pathological feature of several neurodegenerative diseases, including Alzheimer's disease (AD), Parkinson's disease (PD), amyotrophic lateral sclerosis (ALS), and polyglutamine (polyQ) disorders, which among others are referred to as protein misfolding disorders (PMDs) [1]. The aggregation process in PMDs can be triggered by several factors, including genetic mutations, environmental stress, and aging, resulting in the accumulation of oligomeric species which trigger proteostasis imbalance, neuroinflammation, apoptosis, and neuronal degeneration in the brain [2, 3].

Although multiple pathways have been identified as part of the pathophysiological process of PMDs, effective treatments for these diseases are still unavailable. In the last decade, numerous studies have provided associations between the consumption of polyphenolic-rich foods, including berries, and the reduction in the long-term risk to develop neurodegenerative diseases, among other chronic pathologies [4]. *In vitro* studies indicate that polyphenolic compounds, such as flavonoids, can prevent the aggregation of proteins involved in AD [5–7] and PD [8]. Also, polyphenolic compounds have shown protective effects against the toxicity of protein aggregates in cell culture [7, 9–11]. Furthermore, polyphenolic-rich extracts are neuroprotective in animal models of PMDs [12–15]. The possible mechanisms involved in the neuroprotective effects of polyphenols and polyphenolic-rich extracts include direct interactions with key amino acid residues present in misfolded proteins and aggregates [5, 6], the scavenging of reactive oxygen species (ROS) [9], modulation of antioxidant enzymes [9], and the regulation of cellular pathways involved in protein quality control, such as autophagy [16] and the ubiquitin-proteasome system [17]. Thus, intervention strategies based on the use of natural compounds may offer interesting therapeutic avenues to treat neurodegenerative diseases.

HD is the most common polyQ disorder caused by an expansion of the CAG codon in the huntingtin gene (HTT), which causes gain-of-toxic functions of the mutated protein [18]. Expression of mutant HTT disrupts diverse intracellular pathways and leading to the death of medium spiny neurons in the striatum, resulting in progressive loss of motor control and cognitive impairment [19, 20]. Recently, treatment of HD transgenic mice with ellagic acid attenuated neuroinflammation and rescued synaptic dysfunction [11]. Furthermore, the oral administration of an standardized polyphenolic extract obtained from grape seeds resulted in increased lifespan and reduced motor dysfunction of HD mice, correlating with reduced mutant HTT (mHTT) aggregation [14].

*U. molinae* Turcz. (Myrtaceae) is a Chilean native berry shrub commonly known as 'murtilla', whose berries and leaves are a rich source of polyphenolic compounds [21, 22]. The worldwide increment in murtilla berry consumption led to a domestication project conducted by the *Instituto Nacional de Investigaciones Agropecuarias* (INIA) that resulted in the development of 2 commercial varieties of murtilla: Red Pearl and South Pearl [23]. Considering all of the above, our group previously analyzed the possible neuroprotective effects of a phenolic extract obtained from the leaves of wild murtilla using a preclinical mouse model of AD, where it reduced the negative effects of amyloid β aggregates in the brain on hippocampal-dependent cognition and memory [12]. *In vitro* studies also demonstrated potent anti-aggregation activity on a cell free system [12].

The aim of this study was to evaluate and compare the possible protective effects of 16 semi-purified extracts obtained from fruits of 8 different genotypes of murtilla on the aggregation of polyQ peptides and correlate these effects with the chemical composition. Our results uncovered a previously unanticipated anti-aggregation activity of a murtilla berry extract that correlated with its phenolic composition.

## Materials and methods

### Materials

**Chemicals.**    Folin-Ciocalteu reagent, acetonitrile, hexane, acetone, dimethyl sulfoxide (DMSO), ethanol, methanol, gallic acid, and formic acid were purchased from Merck S.A (Darmstadt, Germany). For LC-MS analysis, Milli-Q water was used for the mobile phase in all measurements, and Hibar Purospher Star RP-18 column was purchased from Merck S.A. Cell media and antibiotics were obtained from Invitrogen (MD, USA). Fetal bovine serum was obtained from Hyclone and Sigma. All transfections with plasmids were performed using the Effectene reagent (Qiagen). DNA was purified using columns from Qiagen kits. For the analysis in the cellomics platform, 7-Amino-Actinomycin D (7-ADD) (Thermo Fisher) and Hoescht 33342 stain solutions were used.

**Cell lines and plasmids.**    HEK293 cells were obtained from ATCC and maintained in Dulbecco's modified Eagles medium supplemented with 5% fetal bovine serum and 1× penicillin/streptomycin and maintained at 37˚C and 5% CO2. IMR90 cells were seeded in DMEM-HG with 10% fetal bovine serum and antibiotic/antimycotic and maintained at 37˚C, 10% $CO_2$ and 5% $O_2$. PolyQ$_{79}$-EGFP is a 79 polyglutamine tract in-frame fused to EGFP in the N-terminal; polyQ$_{11}$-EGFP is a 11 polyglutamine tract in-frame fused to EGFP in the N-terminal; mHttQ$_{85}$-GFP contains the first 171 amino acids of the first exon of the huntingtin gene with a tract of 85 glutamines fused to GFP in the N-terminal, and was kindly provided by Dr. Paulson Henry (University of Michigan).

**Plant material.**    Ripe fruits from 8 genotypes of murtilla (*U. molinae* Turcz., Myrtaceae) cultivated under the same climate and soil conditions were collected on April 2016 from the *Instituto Nacional de Investigaciones Agropecuarias* experimental station at Carillanca (INIA Carillanca, Tranapuente, Temuco, Chile). Fruits were stored at -20˚C and lyophilized and triturated before extraction. A voucher sample of each genotype was kept at the herbarium of the *Facultad de Ciencias Químicas y Farmacéuticas of Universidad de Chile*. The number of the selected accessions corresponds to 14–4 (SQF-22549), 19–1 (SQF-22554), 19-1ha (SQF-22553), 22–1 (SQF-22552), 23–2 (SQF-22556), 27–1 (SQF-22555), 31–1 (SQF-22551) and 19–2 (SQF-22557).

**Preparation of extracts.**    Lyophilized and triturated ripe fruits of *U. molinae* were successively and exhaustively extracted by simple maceration at room temperature using solvents of increasing polarity: dichloromethane, acetone, and ethanol (in 1% formic acid), obtaining dicloromethane extracts (DCMEs), acetone extracts (ACEs), and ethanolic extracts (ETEs). Formic acid was added to stabilize any anthocyanin extracted with ethanol. Solvent-to-plant material ratio was 1:5, solvents were changed twice a day, and the maceration was performed until complete exhaustion of the plant material. Complete exhaustion of the plant material was assessed by the loss of color of the solvent being used for extraction. We also performed a qualitative analysis by thin layer chromatography on silica gel GF254 (Merck 5554) and checked for any positive spots under UV light and after spraying with different reagents as Liebermann–Burchard, anisaldehyde and NP/PEG reagent. Once the plant material was completely exhausted with one solvent, it was dried and then macerated with the next solvent. The obtained extracts were collected and concentrated at reduced pressure using a rotatory evaporator and the remaining solvent was evaporated using a vacuum oven at temperatures lower than 70˚C. The solvent-free extracts were stored at -20˚C. ACEs and ETEs were selected for this study because they represent a rich source of phenolic compounds.

## Determination of total phenolic content using the Folin-Ciocalteu spectrophotometric method

Total phenolic content (TPC) of the ACEs and ETEs was determined by the Folin-Ciocalteu (FC) method in a 96-well plate spectrophotometer [21]. Briefly, the ACEs and ETEs were

dissolved in methanol at a concentration of 2.0 mg/mL. The temperature was set at 40˚C and 30 μL of each sample was mixed in a well with 30 μL of FC reagent (Merck; 1:10 in water). After 2 min, 240 μL of 5.0% (w/v) sodium carbonate were added. The final concentration of extract in the plate was 0.2 mg/mL. Absorbance was measured at 765 nm after a 20 min reaction. A calibration curve was prepared with gallic acid (2.0–10.0 mg/mL) as a standard compound (y = 0.0655x + 0.0615, $R^2$ = 0.996) and TPC was expressed as mg gallic acid equivalent (GAE) per g of dry extract (DE).

## Cell viability analysis using automated fluorescence microscopy

Cells were seeded in 96 wells plates (10,000 cells/well) and then treated with increasing concentrations of the extracts in technical duplicates. After 24 h of treatment, cells were incubated with Hoechst 33342, for nuclei staining, and 7-AAD, for dead cells staining, and then imaged on the *Cellomics* VTI Arrayscan using the *General Intensity Measurement* Tool on the HCS Studio® software. Wells treated with DMSO only were used as controls.

## Protein inclusions analysis using automated fluorescence microscopy

Cells were transfected with the polyQ$_{79}$-EGFP peptide and then seeded in 96 well plates (10,000 cells/well). Treatment with the extracts was performed using 4 different concentrations (12.5–100 μg/mL for ETEs and 3.125–25 μg/mL for ACEs) in technical duplicates. After 24 h of treatment, cells were incubated with Hoescht 33342 and nuclei and protein aggregates were imaged on the *Cellomics* VTI Arrayscan using the *General Spot Measurement* Tool on the HCS Studio® software. Non-transfected cells and cells treated with DMSO were used as controls.

## Western blot analysis

HEK293 cells ($3 \times 10^5$) were seeded in 6-well plate and treatment with selected extracts was performed for 16 h unless indicated. For western blot analysis, cells were collected and homogenized in 1% Triton X-100 in PBS containing proteases and phosphatases inhibitors (Roche). After sonication, protein concentration was determined in all experiments by micro-BCA assay (Pierce), and 10–30 μg of total protein was loaded in 8–15% SDS-PAGE minigels (Bio-Rad Laboratories, Hercules, CA) prior transfer into PVDF membranes. Membranes were blocked using PBS, 0.1% Tween-20 (PBST) containing 5% milk for 60 min at room temperature and then probed overnight with primary antibodies in PBS, 0.02% Tween-20 (PBST) containing 5% skimmed milk. The following primary antibodies and dilutions were used: anti-GFP 1:1,000 (Santa Cruz, Cat. n˚ SC-9996), anti-HSP90 1:5,000 (Santa Cruz, Cat. n˚ SC-13119), anti-LC3 1:1,000 (Cell signaling, Cat. n˚ 3868), anti-p62 1:5,000 (Abcam, Cat. n˚ ab56416), Anti-pCHK1(Ser345) 1:1,000 (Cell Signaling, Cat. n˚ CST-2348). Bound antibodies were detected with peroxidase-coupled secondary antibodies incubated for 2 h at room temperature and the ECL system.

## RNA isolation, RT-PCR and real time PCR

HEK293 cells ($3 \times 10^5$) were seeded in 6-well plate and treatment with ETE 19–1 (100 and 200 μg/mL) for 16 h. Quercetin, DMSO and etoposide (Eto) were used as controls. Total RNA was prepared from HEK293 cells using Trizol (Invitrogen, Carlsbad, CA, USA). cDNA was synthesized with SuperScript III (Invitrogen) using random primers p(dN)6 (Roche). Quantitative real-time PCR reactions were performed using SYBRgreen fluorescent reagent and/or Eva-GreenTM using a Stratagene Mx3000P system (Agilent Technologies, Santa Clara, CA 95051,

United States). The relative amounts of human p21 mRNAs were calculated from the values of comparative threshold cycle by using human GAPDH as a Housekeeping.

## Determination of phenolic compounds in the selected extracts (SEs) using mass spectrometry

The selected extracts were analyzed by liquid chromatography coupled to mass spectrometry (LC-MS) using an Agilent 1100 HPLC coupled with an electrospray ion-trap mass spectrometer Esquire 4000 ESI-IT (Bruker Daltonics GmBH, DE) to detect their phenolic compounds profile. For HPLC separation, a Hibar Purospher Star RP-18 column (Waters) with an end-capped of 5 μm and 250 mm x 4 mm was used. The analysis was performed at room temperature by the injection of 20 μL of blank (methanol) or the SEs (10,000 ppm). A gradient system composed by two phases was used according to the methodology described by Pedro et al. (2009): (A) water (MilliQ) and formic acid (4.5%) and (B) acetonitrile. The gradient elution was: 0–22 min 3% B, 22–31 min 22% B, 31–40 85% B, 40–46 min 85% B, 46–56 min 100% B, and 56–65 min 3% B [24]. The flow rate was set at 1.0 mL/min and the UV detection was made at 280 nm. The ionization process (nebulization) was performed at 3,000 V using nitrogen as nebulizer gas at 365˚C, a pressure of 60 psi and a flow rate of 10 L/min. The mass spectra were obtained in negative polarity. All data obtained was analyzed using Bruker DataAnalysis 3.2 (Bruker Daltonik GmbH, DE). Identification of compounds was done by revising the scientific literature [21, 25], employing the ReSpect for Phytochemicals database, and a library from the Unidad de Espectrometría de Masas at Universidad de Chile. The identification was carried out by molecular ion mass and fragmentation pattern comparison. Furthermore, the relative amount of each compound identified was determined. The molecular ion of each compound was selected in the selected ion mode and its area was integrated. For the comparative analysis of the SEs, the highest area of each compound among the SEs was given a 100%.

## Analysis of the chemical profile of the SEs using classical molecular networking

A molecular network was created using an online workflow (https://ccms-ucsd.github.io/GNPSDocumentation/) on the GNPS website (http://gnps.ucsd.edu) [26], which was then exported to Cytoscape for visualization and image obtention. The precursor ion mass tolerance was set to 2.0 Da and a MS/MS fragment ion tolerance of 0.5 Da. A network was then created where edges were filtered to have a cosine score above 0.65 and more than 6 matched peaks. Further, edges between two nodes were kept in the network if and only if each of the nodes appeared in each other's respective top 10 most similar nodes. Finally, the maximum size of a molecular family was set to 100, and the lowest scoring edges were removed from molecular families until the molecular family size was below this threshold. The spectra in the network were then searched against GNPS' spectral libraries. All matches kept between network spectra and library spectra were required to have a score above 0.7 and at least 6 matched peaks.

## Analysis of ETE 19–1 treatment over cell proliferation

HEK293 cells ($3 \times 10^5$) were seeded in 6-well plate and treatment with ETE 19–1 was performed at two concentrations (100 and 200 μg/mL) using quercetin (10 μM) and DMSO as controls. Cells were stained with 0.4% trypan blue and manually counted in duplicates using a Neubauer improved chamber at 0, 24, 48, 72, and 96 h. Alternatively, HEK293 cells were seeded in 96-well plates ($1 \times 10^4$ cells per well) and treated as previously described for 24 and

48 h. Cell number was indirectly determined by the MTS assay following the manufacturer's protocol (Promega, G109A).

## ETE 19–1 effects over cell cycle distribution

HEK293 cells were treated with 100 or 200 μg/ml of ETE 19–1, 10 μM Quercetin, vehicle (DMSO) or left untreated (NT) for 16 h, trypsinized and collected by centrifugation at 1500 rpm for 5 min. The cell pellet was washed with PBS and fixed with 70% ethanol. Cells were treated with 200 μg/ml RNase A (Invitrogen, 12091–021) in annexin-binding buffer (10 mM HEPES, 140 mM NaCl, 2.5 mM CaCl2, pH 7.4) for 2 h, washed in PBS and resuspended in 500 μL of annexin-binding buffer. DNA was stained with 5 μg/ml of propidium iodide (PI) and cells were analyzed by FACS as reported [27]. The phases of the cell cycle were quantified from the PI histogram by multi cycle analysis (FCS express) using the FCS express fit model #6 (CV corrected). 20,000 cells per tube were counted.

## Analysis of ETE 19–1 treatment over endogenous reactive oxygen species (ROS) levels

The generation of intracellular oxidative stress was performed using dihydroethidium (DHE) followed by fluorescence-activated cell sorting (FACS), as previously described [28]. Briefly, HEK293 cells were pre-treated with 100 or 200 μg/ml ETE 19–1, 10 μM Quercetin or vehicle (DMSO) for 16 h and then treated with 500 μM $H_2O_2$ for 2 h. Culture media was replaced with a solution containing 10 μM DHE in HBSS, incubated 20 min at 37˚C, and protected from light. Then, cells were washed in HBSS, trypsinized, resuspended in 200 μL HBSS, and the DHE fluorescence was determined by FACS.

## Analysis ETE 19–1 treatment effects over cellular senescence

IMR90 cells were seeded in DMEM-HG, 10% FBS and antibiotic/antimycotic at CO2 10% and O2 5%. To induce cellular senescence, 5,000 per $cm^2$ IMR90 cells were seeded and the next day, cells were treated with 250 nM doxorubicin for 24 hr. Then, cells were maintained in the plate for 12 days, changing cell culture media every 2 days. To evaluate cellular senescence, SA-B-Galactosidase staining was used, according to the protocol published by Debacq-Chainiaux [29]. Senescent cells were treated with 3 different concentrations of ETE 19–1 (50, 100 and 200 ug/mL) for 24 and 48 hr. ABT 236 was used as control. For cleaved caspase 3 immuno-fluorescence, the cells were fixed with PFA 4% for 10 minutes, wash and then blocked for 60 min in PBS/BSA 10% plus 0.3% Triton™ X-100. After, the primary antibody was applied in 1:50 dilution and incubated overnight at 4˚C. Next day the cells were mounted using Pro-long® Gold Antifade Reagent with DAPI (#8961).

## Statistical analysis

Results are reported as mean ± SD of three independent measurements for spectrophotometric analysis, and as mean ± SEM for cell-based studies. Statistical analyses were carried out using Graph-Pad Prism 6.0 software. Lack-of-fit test, Pearson's correlation (r) test, one-way ANOVA and Tukey's or Dunnet's multiple comparison test were used to analyze the data, considering $p \leq 0.05$ as significant. To evaluate the quality of the polyQ$_{79}$-EGFP inclusion formation assay on the *cellomics* platform of fluorescence microscopy, the Z'-factor coefficient was determined (**Fig 1**). This is a dimensionless factor which is useful for the optimization and validation of high throughput assays [30], and its optimal values go from 0.5 to 1.0.

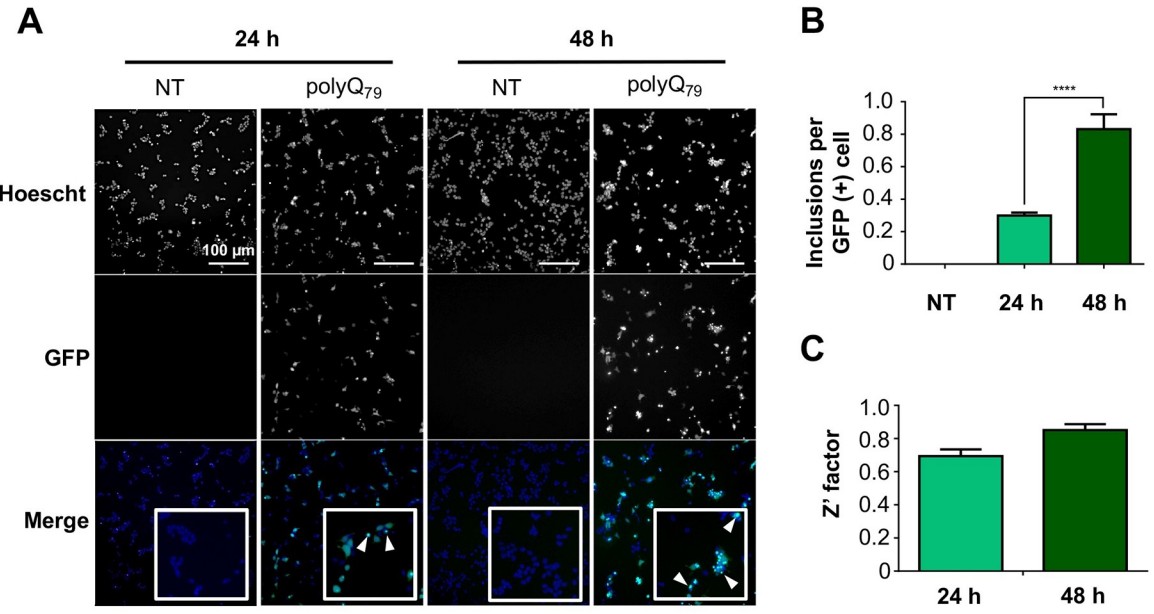

**Fig 1. Analysis of polyQ$_{79}$–EGFP intracellular inclusions and Z′-factor analysis.** Cells were transfected with the polyQ$_{79}$–EGFP plasmid and 24 h after transfection, treated with Hoescht reagent for nuclei staining. Images were acquired at 24 and 48 h after transfection using the *cellomics* platform. GFP-positive (GFP (+)) inclusions were quantified using the *Spot detector* application on the HCS Studio® software. (A) polyQ$_{79}$–EGFP intracellular inclusion formation 24 and 48 h post transfection analyzed using fluorescent microscopy. (B) Quantification of polyQ$_{79}$-EGFP inclusions per GFP (+) cell 24 and 48 h post transfection. **C)** Z' factor analysis 24 and 48 h post transfection. Results are the mean ± SEM of 3 independent experiments. Statistically significant differences were detected by one–way ANOVA (**** = p < 0.0001).

## Results and discussion

### Total phenolic content of semi-purified extracts from different murtilla fruits genotypes

The analysis by the FC method showed significant differences ($p < 0.05$) between ACEs and ETEs. ACEs contained a nearly 4-fold higher total phenolic content (TPC) than ETEs. Also, significant differences ($p < 0.05$) were observed among each set of extracts (**Table 1**). The murtilla genotypes

**Table 1. TPC of the ACEs and ETEs of *U. molinae* fruits from different genotypes assessed by FC method.**

| Total phenolic content ± SD [mg GAE/ g DE] | | |
|---|---|---|
| Genotype | ACEs | ETEs |
| 14–4 | 95.2 ± 6.4 [b,c] | 20.3 ± 1.3 [a] |
| 19–1 | **107.9 ± 4.6** [a,b] | 27.6 ± 1.4 [b,c] |
| 19-1ha | 93.4 ± 4.1 [b,c] | 26.4 ± 0.1 [b] |
| 19–2 | 91.4 ± 8.7 [c] | **32.5 ± 1.6** [c] |
| 22–1 | **111.7 ± 5.0** [a] | 28.2 ± 2.9 [b,c] |
| 23–2 | 93.9 ± 0.5 [b,c] | **30.0 ± 2.0** [b,c] |
| 27–1 | 69.0 ± 1.9 [d] | 23.5 ± 0.4 [a,b] |
| 31–1 | 87.0 ± 2.3 [c,] | 27.4 ± 1.9 [b] |

Different letters represent significant differences (p < 0.05) analyzed by one-way ANOVA and Tukey's multiple comparisons test. TPC = total phenolic content; FC = Folin-Ciocalteu; ACE = acetone extract; ETE = ethanolic extract; GAE = gallic acid equivalents; DE = dry extract.

22–1 and 19–1 presented the highest TPC among ACEs, whereas the 19–2 and 23–2 genotypes showed the highest value among ETEs (**Table 1**). Previously, our group conducted a comparative study on the TPC of the leaves of the different genotypes of murtilla from INIA, Carillanca, also showing significant differences among genotypes [21]. The differences observed in the TPC among the different murtilla samples might be explained by differences in their genotypes, as fruits and leaves were collected from genotypes cultivated under the same climate and soil conditions.

## Extracts from murtilla fruits exert different effects on cell viability

We performed an unbiased cell death assay after 24 h of treatment with different concentrations of ACEs and ETEs using automated fluorescent microscopy to determine a safety range of concentrations for further analysis. Treatment of cells with ACEs resulted in increased cell death when compared with ETEs, reaching around 80% of toxicity at the highest concentration tested (100 µg/mL; **Fig 2A**). Meanwhile, all ETEs showed less than 5% of cell death (**Fig 2A**). This differences in cell viability are in accordance with the differences observed in the TPC results (**Table 1**), where ACEs showed a significantly higher TPC value. Although phenolic compounds are present in several herbal formulations, commercial extracts, foods and beverages, and their consumption is generally considered safe and beneficial for human health, toxicity assessment represents a key step in the drug development process [31]. In this context, it

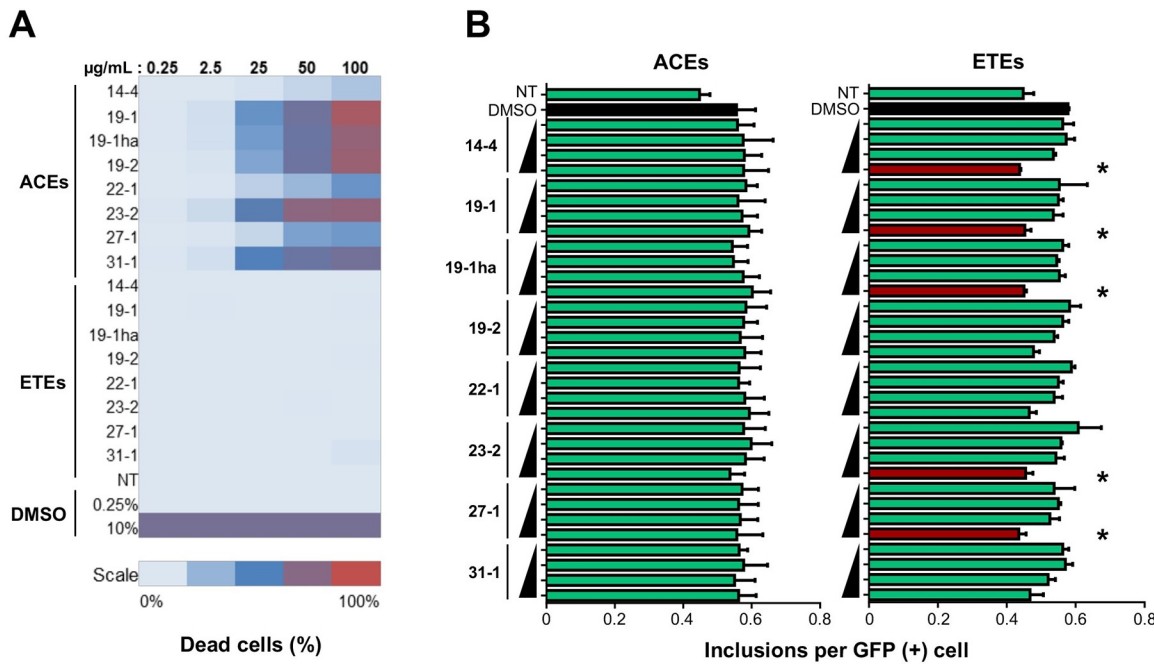

**Fig 2. Treatment with extracts from different genotypes of *U. molinae* fruits exerts differential effects on cell viability and the number of polyQ$_{79}$-EGFP inclusions in HEK293 cells.** The analysis was performed in the *cellomics* platform of automated fluorescence microscopy. (A) For the cell viability analysis, cells were seeded and then treated with increasing concentrations of each extract. After 24 h, cells were treated with 7-AAD and images were acquired with the *cellomics* platform. 7-AAD-stained cells were quantified using the *General intensity measurement* tool. Data was acquired using the HCS Studio® software. (B) For the analysis of the effects over polyQ$_{79}$-EGFP inclusions, cells were transfected and then treated with increasing concentrations of each extract. A range of concentrations from 3.1 to 12.5 µg/mL were analyzed for ACEs, except for ACE 14–4 (12.5 to 100 µg/mL). For ETEs, concentrations from 12.5 to 100 µg/mL were analyzed. After 24 h, images were acquired in the *cellomics* platform and GFP (+) protein inclusions were quantified using the *spot detector* application. Data was acquired using the HCS Studio® software. Results are the mean ± SEM of 3 independent experiments. Statistically significant differences were detected by one-way ANOVA and Tukeys multiple comparisons considering DMSO as control (* = p < 0.05).

has been reported that treatments with high concentrations of phenolic-rich extracts exert cytotoxicity *in vitro*, which could be partially explained by the pro-oxidant and pro-mutagenic effects that polyphenolics can exert at high concentrations [32]. Considering these results, ACEs concentrations for further analysis were adjusted to 3.1–25.0 μg/mL with exception of ACE 14–4, that as ETEs, was analyzed at concentrations between 12.5 and 100 μg/mL.

## A murtilla berry extract decreases polyQ$_{79}$-EGFP and mHTT-GFP levels in cellular models of HD

First, to evaluate the quality of the polyQ$_{79}$-EGFP inclusion formation in our cellular model, we transfected HEK293 cells with polyQ$_{79}$-EGFP and seeded the cells in a 96-well plate. 24 and 48 h post transfection, inclusion formation was analyzed using automated fluorescence microscopy and the Z'-factor was calculated for the assay. Optimal values (Z' > 0.5) were obtained, meaning that the conditions for polyQ$_{79}$-EGFP inclusion formation were optimal (**Fig 1**). Then, we evaluated the possible impact of our extracts in the load of polyQ$_{79}$-EGFP inclusions in a HD cellular model. HEK293 cells were transfected and then treated with different concentrations of ACEs and ETEs, as previously described, followed by automatic visualization and quantification of GFP-positive puncta after 24 h. ACEs treatments did not modify the number of polyQ$_{79}$-EGFP inclusions per cell (**Fig 2B, left panel).** On the other hand, although having a lower TPC compared to ACEs, treatment of cells with the highest concentration of the ETEs (100 μg/mL) obtained from the 14–4, 19–1, 19-1ha, 23–2 and 27–1 genotypes produced a significant decrease ($p < 0.05$) in the number of polyQ$_{79}$-EGFP inclusions (**Fig 2B, right panel**). The stronger beneficial effect was observed with the ETE 27–1 (South Pearl), which was able to reduce polyQ$_{79}$-EGFP inclusions ~ 25%. Furthermore, ETE 14–4 treatment led to approximately 24% reduction of inclusion content, ETE 19-1ha resulted in a 22% reduction, whereas ETE 19–1 (Red Pearl) and ETE 23–2 presented a 21% reduction. The differences in the activity of the extracts were not directly correlated to their TPC values (**Table 1 and S1 Fig**), suggesting that their effects may be related to their polyphenolic profile or a specific combination of these molecules.

Five ETEs were selected (SEs) for further secondary validation using standard biochemical approaches. For these experiments, HEK293 cells were transfected with polyQ$_{79}$-EGFP vector or polyQ$_{11}$-EGFP vector, used as control. Then, cells were treated with 100 μg/mL of each SEs and 16 h later lysed to generate protein extracts for western blot analysis. Among the 5 extracts evaluated, only ETE 19–1 (Red Pearl variety) treatment showed a significant ($p < 0.05$) reduction of high molecular weight (HMW) species of polyQ$_{79}$- EGFP levels by up to 50% compared to control (**Fig 3A and 3B**). This effect was also observed in the polyQ$_{79}$-EGFP monomer levels (**Fig 3A–3C**), but not over polyQ$_{11}$-EGFP levels, which does not form intracellular inclusions (**Fig 3D and 3E**), suggesting that ETE 19–1 effects are against the misfolded protein. Moreover, these results demonstrate that murtilla fruit ETEs treatment can decrease polyQ$_{79}$-EGFP aggregation levels in a cellular model that recapitulates one of the central hallmarks of HD, the pathological accumulation of misfolded proteins inside the cell [19].

HD pathology is characterized by the generation of a N-terminal fragment of HTT which is highly susceptible to aggregation and formation of nuclear inclusions in striatal neurons [33]. Thus, we then evaluated the effects of the SEs on the aggregation of a construct corresponding to exon 1 of the huntingtin protein containing the first 171 amino acids with a tract of 85 glutamines fused to GFP in the N-terminal region (here termed mHTT$_{Q85}$-GFP). Remarkably, we observed that the ETE 19–1 (Red Pearl) extract showed the strongest anti-aggregation effect, reducing the content of mHTT$_{Q85}$-GFP HMW species by nearly 50% (p < 0.05), followed by the ETE 27–1 (South Pearl) with a reduction of HMW species levels around 35%, but this

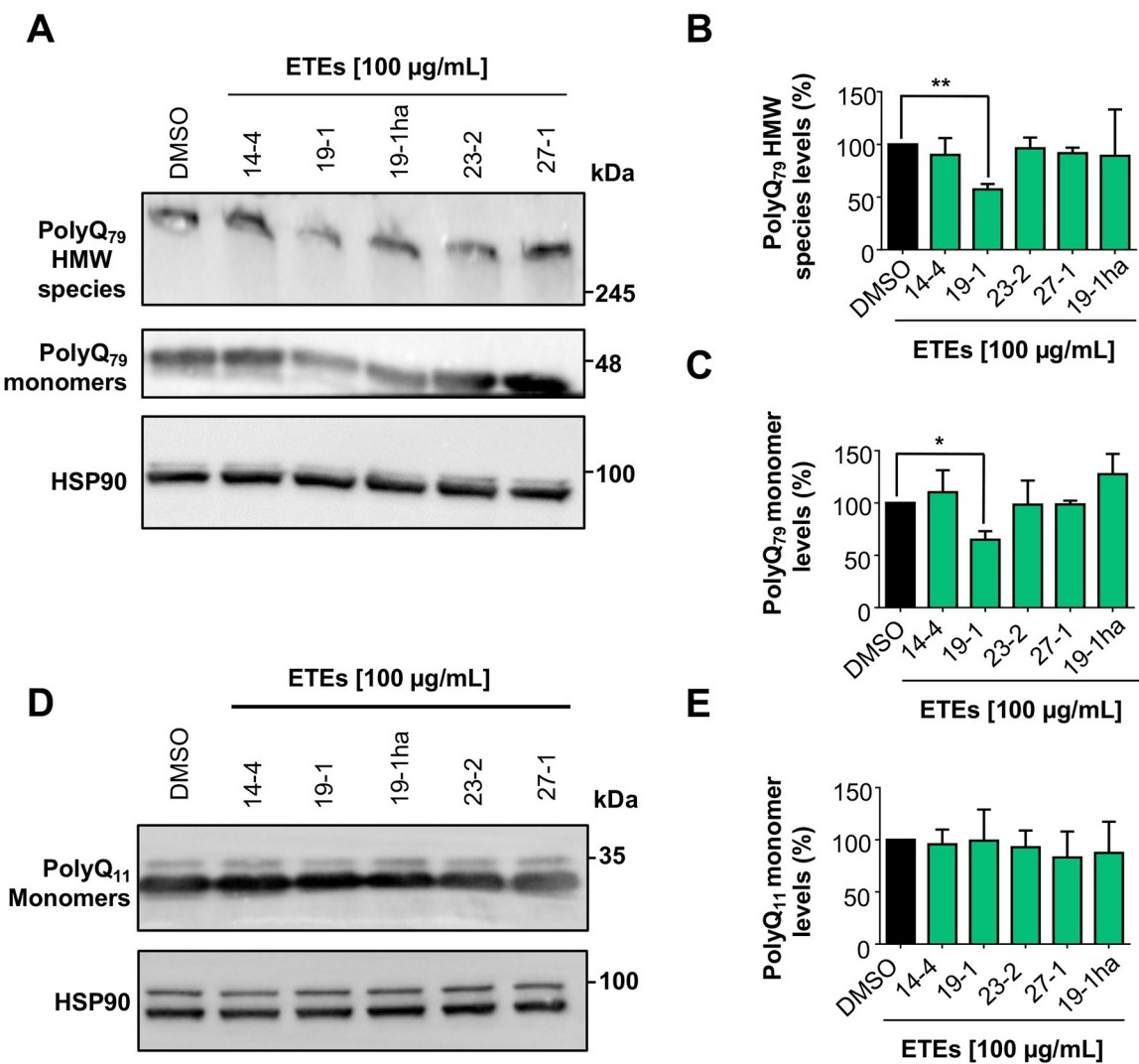

**Fig 3. ETE of *U. molinae* fruits from the 19–1 genotype decreases the levels of polyQ79-EGFP in HEK293 cells.** (A) The effect over polyQ79-EGFP levels was analyzed by western blot. Cells were transfected with the polyQ79-EGFP and 4 h post-transfection, cells were treated with the SEs at a 100 μg/mL concentration. After 16 h, cells were collected, and total proteins extracted. PolyQ79-EGFP monomer and HMW species levels were analyzed using an anti-GFP antibody. HSP90 expression was monitored as loading control. (B) PolyQ79-EGFP HMW species quantification. (C) PolyQ79-EGFP monomer quantification. (D) The effect over polyQ11-EGFP levels was analyzed by western blot as described before. PolyQ11-EGFP monomer levels were analyzed using an anti-GFP antibody. HSP90 expression was monitored as loading control. (E) PolyQ11-EGFP monomer quantification. Results are the mean ± SEM of 3 independent experiments. Statistically significant differences were detected by one-way ANOVA and Dunnett's multiple comparisons test considering DMSO as control (* = p < 0.05; ** = p < 0.01).

reduction did not reach statistical significance (p = 0.0708) (**Fig 4A and 4B**). A similar tendency was observed for ETE 23–2 with a 23% reduction (p = 0.2620) (**Fig 4A and 4B**). No significant reduction in the mHTT$_{Q85}$-GFP monomer levels was observed after treatment with the SEs (**Fig 4A–4C**). These results are promising since the N-terminal fragment of HTT is highly pathogenic and its aggregation is involved in the neurodegenerative process described in HD [33].

HD is also characterized by a functional decline of autophagy that contributes to the abnormal accumulation and aggregation of proteins [34]. Therefore, the autophagic machinery

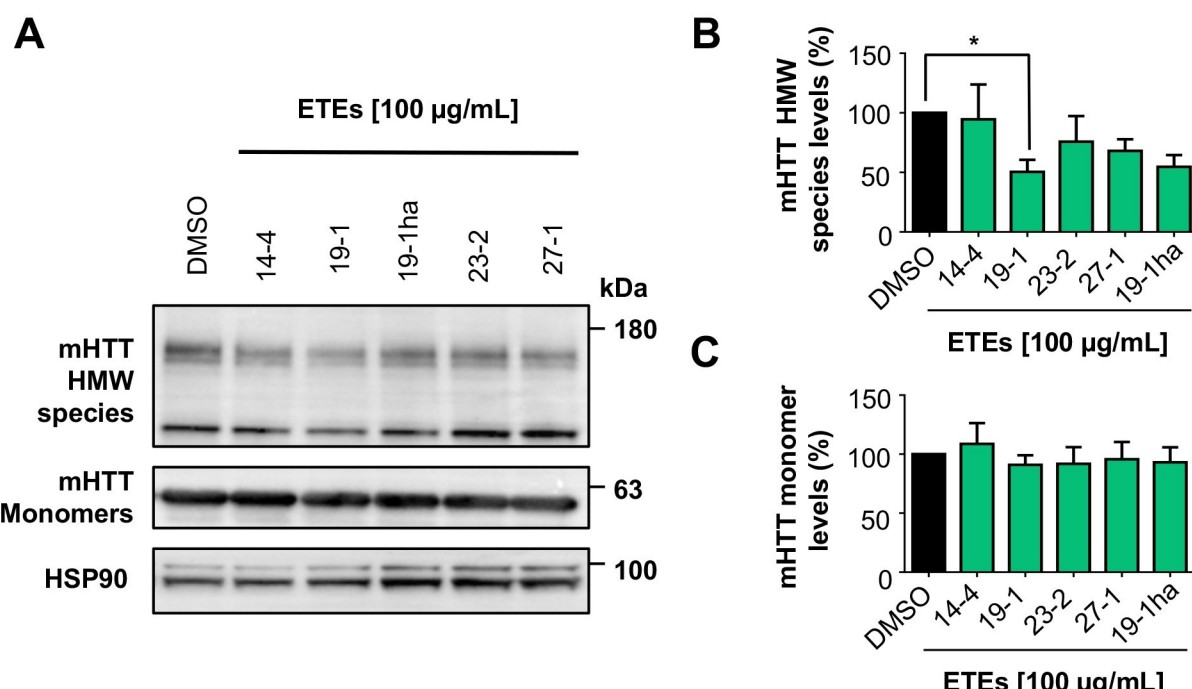

**Fig 4. ETE 19–1 significantly reduces the levels of mHTT$_{Q85}$-GFP levels in HEK293 cells.** (A) The effect over mHTT$_{Q85}$-GFP HMW species and monomer levels were analyzed by western blot. Cells were transfected with the plasmid that encodes for the first 171 amino acids of HTT with an 85 glutamines extension. 4 h post transfection, cells were treated with each of the SEs at a 100 μg/mL. 16 h later, cells were collected, and total proteins extracted. mHTT$_{Q85}$-GFP HMW species and monomer levels were analyzed using an anti-GFP antibody. HSP90 expression was monitored as loading control. (B) mHTT$_{Q85}$-GFP HMW quantification. (C) mHTT$_{Q85}$-GFP monomer quantification. Results are the mean ± SEM of 3 independent experiments. Statistically significant differences were detected by one-way ANOVA and Dunnett's multiple comparisons test considering DMSO as control (* = $p < 0.05$).

represents an interesting target for HD treatment. Pharmacological and genetic approaches that modulate autophagy have shown promising results in diverse HD models [35, 36]. Moreover, autophagy induction by natural products, and more specifically polyphenolic compounds, has shown to be protective in *in vitro* [37] and *in vivo* [11]. Thus, we analyzed the effect of ETE 19–1 treatment on the levels p62 and the lipidated form of LC3 (LC3-II) (**Fig 5**), as the modulation of the levels of these proteins is correlated with the induction of autophagy [38]. After ETE 19–1 treatment in cells overexpressing polyQ$_{11}$ and polyQ$_{79}$ we observed higher p62 (**Fig 5B**) and LC3-II (**Fig 5C**) levels compared to DMSO-treated cells, although this increment was not significant ($p > 0.05$) in any condition. The higher levels of p62 and LC3 compared to control reflect a functional induction of the autophagy process by ETE 19–1, which in turn might be related to its effects on protein aggregation.

## The composition of SEs is rich in flavonoid glycoside derivatives

Finally, we evaluated the phenolic profile of the SEs in order to identify potential compounds that could mediate their effects on protein aggregation. The comparative identification of phenolic compounds in the SEs was analyzed using LC-MS. In total, 104 compound signals were evaluated, and 102 molecules were tentatively identified using as reference a library of phenolic compounds, on-line databases (MassBank, ReSpect for Phytochemicals) and available literature [21, 25]. Furthermore, a comparative semi-quantification of the evaluated signals was performed to analyze the relative phenolic content and composition among the SEs. These results

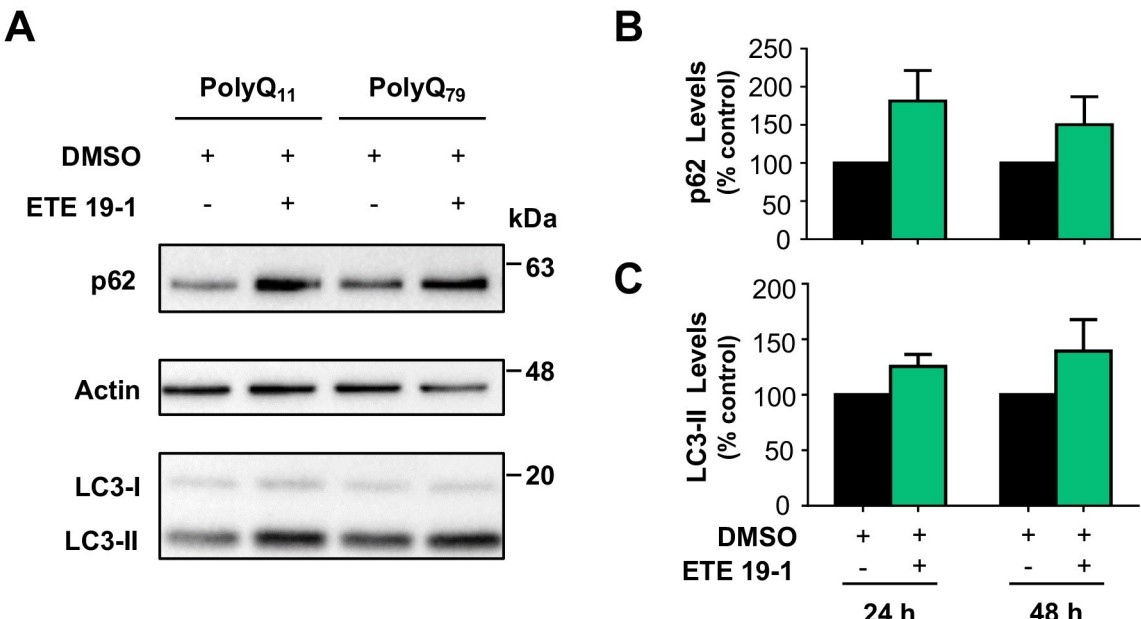

**Fig 5. ETE 19–1 modulates the levels of autophagy-related proteins.** (A) Modulation of autophagy was assessed by western blot analyzing the levels of p62 and LC3-II. Cells were transfected with polyQ$_{11}$-EGFP, polyQ$_{79}$-EGFP, or mHTT$_{Q85}$-GFP expression vectors. 4 h post transfection, cells were treated with ETE 19–1 at 100 μg/mL. 16 h later, cells were collected, and total protein extracted. p62 and endogenous lipidated LC3-II levels were monitored by western blot using anti-p62 and anti-LC3 antibodies, respectively (upper and lower panel). Actin expression was monitored as loading control (middle panel). (B) p62 and (C) LC3-II levels were quantified and normalized to actin. Results are the mean ± SEM of 3 independent experiments. Statistically significant differences were detected by unpaired $t$-test.

are shown in **S1 Table**. To further analyze the chemical profile of the SEs, we used the GNPS platform to create a molecular network that clusters related chemical compounds by similarities in their fragmentation patterns (**Fig 6**), allowing us to create a visual display of the chemical profile of the SEs. In the molecular network, each node represents a precursor ion and the area of each color in the nodes represents the abundance of the precursor in each extract (**Fig 6**).

Most of the identified compounds in the SEs correspond to glycoside derivatives of quercetin and myricetin. It was also possible to identify the presence of glycoside derivatives of other flavonoids as kaempferol and isorhamnetin, ellagitannins as HHDP-galloyl hexose, and phenolic acids as caffeic acid derivatives in the SEs (**S1 Table**). Although the analyzed extracts were obtained from fruits of different genotypes, the phenolic profiles of these samples were similar (**Figs 6 and 7**). Interestingly, only ETE 19–1 had significant effects over both polyQ$_{79}$ and mHTT aggregation (**Figs 3 and 4**) thus, a specific mixture of phenolic compounds in this extract might be related to its effects over protein aggregation. This is supported by the semi-quantification analysis, which showed a clear difference in the relative quantity of the phenolic compounds identified among the SEs (**S1 Table**), and the molecular network created using the GNPS platform, where compounds that were only detected in the ETE 19–1 are showed as fully red nodes (**Fig 6**). Although the presence of other types of phytoconstituents cannot be discarded, the sequential extraction process used in our study allowed us to obtain ETEs that were enriched in polyphenolic compounds (Figs 6 and 7), thus reducing the influence of other compounds over their pharmacological properties.

Many studies have identified the beneficial effects of phenolic compounds, especially of flavonoids, over protein aggregation related to neurodegeneration. *In vitro* studies indicated that

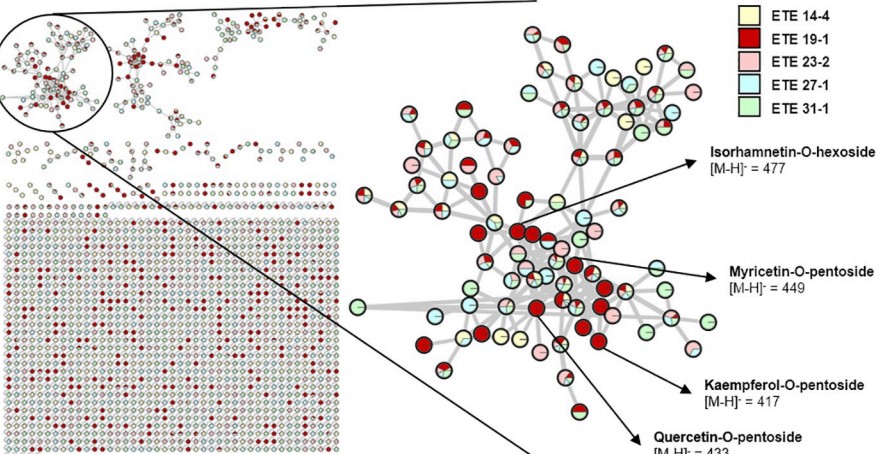

**Fig 6. Molecular network of the 5 selected ethanolic extracts (SEs) of *U. molinae* fruits from different genotypes.** The molecular network was created using the online workflow (https://ccms-ucsd.github.io/GNPSDocumentation/) on the GNPS website (http://gnps.ucsd.edu) and then exported to Cytoscape. Each node represents a precursor ion and clusters of related molecules were formed by analyzing the similarities among their fragmentation patterns. The area of each color in the nodes represents the abundance of the precursor in each extract. One cluster containing flavonol glycoside derivatives is shown in detail and some of the compounds identified only in ETE 19–1 are presented.

flavonoid derivatives, like the ones identified in our murtilla extracts (i.e., quercetin, myricetin and kaempferol), could inhibit the formation and extension of amyloid β fibrils, as well as to destabilize preformed fibrils, in a dose-dependent manner [6, 7]. Similar anti-aggregation effects have been reported using recombinant α-synuclein fibrils [8]. Also, treatment of cells with polyphenolic compounds, as well as polyphenolic-rich plant extracts, provide protection against the toxicity elicited by protein aggregates in cellular models of AD [9], ALS [10], and HD [11]. Positive effects have been also reported after the oral administration of phenolic-rich extracts to animal models of PMDs. For example, treatment of transgenic mouse models of AD and HD with a phenolic-rich standardized extract obtained from grape seeds reduces protein aggregation in the brain, attenuating cognitive deterioration [39] and motor skill decline [14] respectively. Furthermore, using a preclinical mouse model of AD, our group demonstrated that the treatment of AD mice with a phenolic extract obtained from wild murtilla leaves reduced amyloid β plaque formation in the brain, which translated into improvements of cognitive functions of treated animals [12].

Our results show that autophagy induction might be involved in the decrease of protein aggregates in cells by the ETE 19–1 extract (**Fig 5**). Compounds that are present in this extract, like quercetin glycosides and ellagic acid, can engage autophagy and protect against the deleterious effects of toxic protein aggregates in *in vivo* models of HD [11, 40], supporting our results. It is also feasible to speculate that compounds present in ETE-19-1 directly destabilize protein aggregates or activate other cellular pathways that mediate protein aggregates degradation. In agreement with this idea, we have observed that murtilla leaf extracts are able to inhibit aggregation and disaggregate preformed amyloid β fibrils on a cell-free system *in vitro* [41]. At the chemical level, the presence of aromatic rings and vicinal hydroxyl groups, as the ones in catechol groups, are proposed to direct the interaction of flavonoids with key amino-acid residues in proteins related to neurodegeneration by hydrogen bonding and/or covalent interactions [5, 6]. Alternatively, murtilla fruit extracts may induce the degradation of toxic protein species by the activation intracellular clearance pathways, like the proteasome, or by engaging

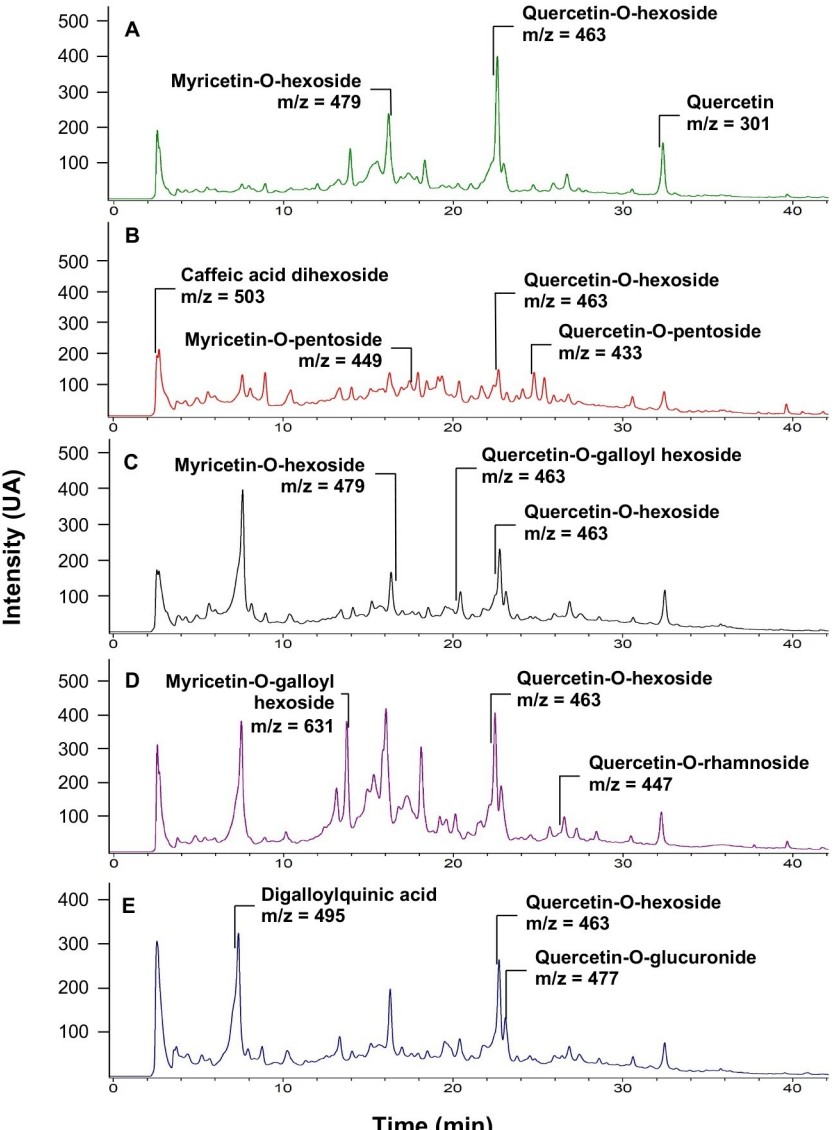

**Fig 7. LC-UV chromatograms (280 nm) of the 5 selected ethanolic extracts (SEs) of *U. molinae* fruits from different genotypes obtained from the LC-MS/MS analysis.** The LC-MS/MS data was analyzed using Bruker DataAnalysis 3.2 software (Bruker Daltonik GmbH, DE). Tentative identification of phenolic compounds was carried out by molecular ion mass and fragmentation pattern comparison with a library and the ReSpect *for* Phytochemicals database. Some of the compounds detected in each extract are shown. (A) ETE 14–4 LC-UV chromatogram. (B) ETE 19–1 LC-UV chromatogram. (C) ETE 19-1ha LC-UV chromatogram. (D) ETE 23–2 LC-UV chromatogram. (E) ETE 27–1 LC-UV chromatogram.

quality control mechanisms (i.e. the unfolded protein response or the integrated stress response) [42]. Thus, ETE 19–1 may have a dual role in protein aggregation associated with PMDs, as it might directly destabilize protein aggregates and induce intracellular mechanisms to cope with protein misfolding.

It has been described that polyphenolic compounds and polyphenolic-rich extracts can modulate cell proliferation, cell cycle, and endogenous ROS levels [43, 44], which could also have an impact on the observed effects over protein aggregation. We analyzed the effects of ETE 19–1 treatment on cell proliferation my counting cells with trypan blue or by using the

MTS assay. Treatment of cells with quercetin was also included. No significant changes were observed in cell proliferation in cells treated with two concentrations of ETE 19–1 (see methodological details in S2A and S2B Fig). Similarly, no alteration in cell cycle were detected using after the analysis of DNA content using propidium staining of permeabilized cells followed by FACS analysis (S2D Fig). Basal ROS level were monitored using the fluorescent probe dihydroethidium and FACS analysis, observing no effects of ETE 19–1 (S2E Fig). As positive control cells were treated with $H_2O_2$. Furthermore, ETE 19–1 treatment did not induce DNA damage on HEK293 cells as monitored by the levels of phosphorylated CHK1 and the upregulation of p21 mRNA (S3 Fig).

Senescence is a cellular process that is activated in stressed cells to prevent the proliferation of damaged cells [45]. The accumulation of senescent cells has been indicated as a potentially important factor contributing to aging and aging-related diseases, as PMDs [46]. Protein aggregation has been associated to cellular senescence of the brain [47] and senolytics, compounds that can clear senescent cells, have shown to alleviate cognitive deficits in AD models [48]. Quercetin, one of the polyphenolic compounds that is present in ETE-19, has been indicated as a senolytic compound [49, 50], but reports are controversial as other studies indicate that quercetin does not act as a senolytic in human endothelial cells [51] and fisetin, a quercetin derivative, might act as a senolytic or senomorphic depending on the cell type [45]. Nevertheless, we analyzed the effects of ETE 19–1 treatment on senescence and observed that this polyphenolic-rich extract did not have a senolytic effect (S4 Fig). In consequence, the modulation of cellular senescence may not contribute to the effects of ETE 19–1 on protein aggregation.

The bioavailability of these compounds is also important to consider, as polyphenolic compounds might not reach the brain after oral consumption. The fact that the oral treatment with polyphenolic compounds or polyphenolic-rich extracts in animal models of PMDs has shown beneficial effects [12, 14] might be explained by the availability of orally-ingested polyphenols in the brain. Furthermore, the plasma bioavailability of different polyphenols after berry consumption in humans has been confirmed [52] and the mechanisms through which these compounds cross the blood brain barrier to reach the brain have been studied [53].

Overall, our results demonstrate a protective effect of the phenolic-rich ETE 19–1 extract over abnormal protein aggregation associated with neurodegenerative conditions (**Figs 2B–4**) which might be mediated through autophagy activation (**Fig 5**). Considering the low levels of toxicity shown by all ETEs (**Fig 2A**) and the effects on protein aggregation, the follow-up studies should focus on testing fractions of ETE 19–1 through a bio-guided chemical study in order to identify the molecules with anti-aggregation activity and administrating the ETE 19–1 to animal models of neurodegeneration to corroborate its effects. Furthermore, synergism or summation between the different compounds that are present in this extract (**Figs 6 and 7**) may motivate the direct use of ETE 19–1 in further studies to evaluate mutant huntingtin aggregation, neuroprotection, and life span in preclinical models of HD and other PMDs models.

## Conclusions

In this study we have identified significant differences in the neuroprotective properties of ACEs and ETEs obtained from *U. molinae* fruits of different genotypes. Among those extracts, ETE 19–1 extracts had a potent activity over the abnormal protein aggregation on two different cellular models of HD, which might be mediated, in part, by the modulation of the proteostasis network (i.e. autophagy induction). The differences observed in the biological activities of the ETEs on protein aggregation might be explained by variations found in their polyphenolic

compound profile, where glycosylated derivatives of quercetin, myricetin, and kaempferol, alongside phenolic acids and ellagitanins were identified. Interestingly, the polyphenolic profile of the different extracts is similar, thus the effects of ETE 19–1 over protein aggregation might be explained by a specific mixture of compounds in the extract. Further analysis of the effects of ETE 19–1 polyphenolic components on protein aggregation needs to be performed in order to identify lead compounds responsible for its aggregation-lowering effects and their mechanisms of action. Considering all the results, the differences observed might be explained by the variances in murtilla fruits genotypes. Thus, a novel and unique phenolic-rich and non-toxic extract obtained from *U. molinae* fruits, a native Chilean berry, has promising effects on reducing protein aggregation related to neurodegenerative diseases. Hence, ETE 19–1 represents an interesting candidate for further studies in protein misfolded disorders using *in vivo* models. Finally, this study provides additional evidence supporting a the beneficial effect of berry consumption to improve human health where murtilla berries arise as a potential nutraceutic to prevent the accumulation of toxic protein species.

## Supporting information

**S1 Table. Tentative identification and relative quantification of phenolic compounds in the SEs of *U. molinae* fruits from different genotypes.** SE = Selected extract; Rt = retention time. Semiquantification was carried out in respect to the higher normalized area obtained from selective ion mode in LC-MS analysis. n.i. = not identified. [a]CEPEDEQ library; [b]ReSpect for Phytochemicals Database; [c]Reference: Peña-Cerda et al., 2017 [24]; Wyrepkowski et al., 2014 [27].
(DOC)

**S1 Fig. The TPC is not correlated to the effect of the SEs on the number of polyQ$_{79}$-EGFP intracellular inclusions.** To analyze the correlation between the TPC of the SEs and their effects over intracellular polyQ$_{79}$-EGFP inclusions a linear regression between both sets of data was performed.
(DOC)

**S2 Fig. ETE 19–1 treatment does not affect cell proliferation, cell cycle, or induce changes in basal endogenous ROS levels.** (A) TPC of ETE 19–1 was assessed by FC method to check for major changes in its phenolic composition during time. (B) HEK293 cells were seeded in 6-well plate and treatment with ETE 19–1 was performed at two concentrations (100 and 200 μg/mL) using quercetin (10 μM) and DMSO as controls. Cells were stained with 0.4% trypan blue and manually counted in duplicates using a Neubauer improved chamber at 0, 24, 48, 72, and 96 h. (C) HEK293 cells were seeded in 96-well plates (1x104 cells per well) and treated as previously described for 24 and 48 h. Cell number was indirectly determined by the MTS assay. (D) HEK293 cells were treated with 100 or 200 μg/ml of ETE 19–1, 10 μM Quercetin, vehicle (DMSO) or left untreated (NT) for 16 h. DNA was stained with 5 μg/ml of propidium iodide (PI) and cells were analyzed by FACS. (E) HEK293 cells were pre-treated with 100 or 200 μg/ml ETE 19–1, 10 μM Quercetin or vehicle (DMSO) for 16 h and then treated with 500 μM H2O2 for 2 h. Culture media was replaced with a solution containing 10 μM DHE in HBSS, incubated 20 min at 37˚C, and protected from light. Then, cells were washed in HBSS, trypsinized, resuspended in 200 μL HBSS, and the DHE fluorescence was determined by FACS. Results are reported as mean ± SD of three independent measurements for spectrophotometric analysis, and as mean ± SEM for cell-based studies. Statistical analyses were carried out using Graph-Pad Prism 6.0 software. T-test, one-way ANOVA and Tukey's or Dunnet's multiple comparison test were used to analyze the data, considering p ≤ 0.05 as significant.

NT = not treated; DMSO = dimethyl sulfoxide; GAE = gallic acid equivalents; DE = Dry extract; DHE = Dihydroethidium; MFI = mean fluorescence intensity.
(DOC)

**S3 Fig. ETE 19–1 treatment does not induce DNA damage.** (A) The effect of ETE 19–1 treatment over pCHK1 expression was analyzed by Western Blot. HEK293 cells were seeded in 6-well plate and treatment with ETE 19–1 (100 and 200 μg/mL) for 16 h. Quercetin, DMSO and etoposide (Eto) were used as controls. After 16 h, cells were collected, and total proteins extracted. pCHK1 levels was analyzed using an Anti-pCHK1 antibody. HSP90 expression was monitored as loading control. (B) The effect of ETE 19–1 treatment over p21 was analyzed by Quantitative real-time PCR. HEK293 cells were seeded in 6-well plate and treatment with ETE 19–1 (100 and 200 μg/mL) for 16 h. Quercetin, DMSO and etoposide (Eto) were used as controls. Total RNA was prepared from HEK293 cells using Trizol and cDNA was synthesized with SuperScript III (Invitrogen) using random primers p(dN)6. Quantitative real-time PCR reactions were performed using SYBRgreen fluorescent reagent and/or EvaGreenTM using a Stratagene Mx3000P system. The relative amounts of human p21 mRNAs were calculated from the values of comparative threshold cycle by using human GAPDH as a Housekeeping. Results are reported as mean ± SEM of three independent measurements. Statistical analyses were carried out using Graph-Pad Prism 6.0 software. One-way ANOVA and Dunnet's multiple comparison test were used to analyze the data, considering $p \leq 0.05$ as significant.
NT = not treated; DMSO = dimethyl sulfoxide; Q = Quercetin; Eto = Etoposide.
(DOC)

**S4 Fig. ETE 19–1 treatment does not induce death by apoptosis in senescent cells.** Senescence was induced in IMR90 cells as described in the methods section. Senescent cells were treated with ETE 19–1 and 3 concentrations were tested (50, 100, and 200 μg/mL). (**A**) The presence of cleaved caspase 3 was evaluated by immunofluorescence at 0, 24, and 48 hrs. 100 cells were counted and ABT 263 was used as a control. Cell nuclei can be seen in blue (Hoetsch) and cleaved caspase 3 can be seen in green. (**B**) Quantification of cleaved caspase 3 positive cells. Results are the mean of 2 experiments.
(DOC)

**S5 Fig. Raw images.**
(DOC)

## Acknowledgments

The authors would like to thank Instituto Nacional de Investigaciones Agropecuarias (INIA, Carillanca, Chile) for the plant material and their help, and SouthAm freeze dry for the lyophilization of the plant material.

## Author Contributions

**Conceptualization:** Rodrigo Pérez-Arancibia, Jose Luis Ordoñez, Philippe Pihán, Alfredo Sagredo, Ulises Ahumada, César Cárdenas, Rene L. Vidal, Claudio Hetz, Carla Delporte.

**Data curation:** Rodrigo Pérez-Arancibia.

**Formal analysis:** Rodrigo Pérez-Arancibia.

**Funding acquisition:** Rodrigo Pérez-Arancibia, Jose Luis Ordoñez, Alexis Rivas, Rene L. Vidal, Claudio Hetz, Carla Delporte.

**Investigation:** Rodrigo Pérez-Arancibia, Philippe Pihán, Alfredo Sagredo, Ulises Ahumada.

**Methodology:** Rodrigo Pérez-Arancibia, Jose Luis Ordoñez, Alexis Rivas, Andrés Barriga.

**Project administration:** Claudio Hetz, Carla Delporte.

**Resources:** Andrés Barriga, Ivette Seguel.

**Supervision:** Rene L. Vidal, Claudio Hetz, Carla Delporte.

**Writing – original draft:** Rodrigo Pérez-Arancibia.

**Writing – review & editing:** Rodrigo Pérez-Arancibia, Rene L. Vidal, Claudio Hetz, Carla Delporte.

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
