## [Decision Letter · Decision Letter 0]

23 Feb 2021

PONE-D-21-01842

A phenolic-rich extract from Ugni molinae berries reduces abnormal protein aggregation in a cellular model of Huntington’s disease

PLOS ONE

Dear Dr. Delporte,

Thank you for submitting your manuscript to PLOS ONE. After careful consideration, we feel that it has merit but does not fully meet PLOS ONE’s publication criteria as it currently stands. Therefore, we invite you to submit a revised version of the manuscript that addresses the points raised during the review process.

Several methodological concerns are stated in the Reviewers' reports and have to be meticulously addressed during revising. Please provide point-by-point response to Reviewers and if a particular point needs discussion, please offer one. The Discussion section should be considerably deepened in the directions provided by the Reviewer #2. 

We look forward to receiving your revised manuscript.

Kind regards,

Branislav T. Šiler, Ph.D.

Academic Editor

PLOS ONE

Reviewers' comments:

Reviewer's Responses to Questions

**Comments to the Author**

1. Is the manuscript technically sound, and do the data support the conclusions?

Reviewer #1: Yes

Reviewer #2: Yes

2. Has the statistical analysis been performed appropriately and rigorously? 

Reviewer #1: Yes

Reviewer #2: N/A

3. Have the authors made all data underlying the findings in their manuscript fully available?

Reviewer #1: Yes

Reviewer #2: Yes

4. Is the manuscript presented in an intelligible fashion and written in standard English?

Reviewer #1: Yes

Reviewer #2: Yes

5. Review Comments to the Author

Reviewer #1: The present study has studied the effect of phenolic rich extract of Ugni molinae berries against protein aggregation in neurodegenerative disorders especially HD. The study has been very well designed and all the methods used in this study have been explained in sufficient details. The results and discussion of the present manuscript is also very well articulated and explicitly explained. Following are some of my observations which require some clarity:

1. Authors have used ‘simple maceration’ for extracting the plant material. What was the duration of extraction and also mention the solvent to material ratio as these are important parameters that will affect the extraction efficiency. Since the effect of extracts is correlated with the phenolic content, these parameters are important for reproducibility of the results.

2. Why 1% formic acid in ethanol was used for extraction?

3. Authors have stated that polyQ79-EGFP inclusions are not correlated with the TPC of different ETEs but with polyphenolic content. How you differentiate between total phenolic and polyphenolic content? Further, ETE19-2 exhibited highest phenolic content, what was its effect on polyQ79-EGFP inclusions?

4. The results of the extracts have been mainly correlated with the phenolic compound especially, flavonoid glycosides. But the effects of different extracts are not in concordance with the TPC. Authors have mentioned that, the effects of ETE 19-1 are attributed to a specific combination of phenolic compounds present in the extract. Was the ETE free from other type of phytoconstituents? Since constituents belonging to other phytochemical class are also soluble in ethanol, there may be possibility that phytoconstituents belonging to other classes might play a role in the activity. If yes then this should also be mentioned and not only the phenolic compounds.

5. The only limitation of the study is the correlation of the activity of the berries seed extract with polyphenolic compounds only on basis in-vitro studies only. This should be mentioned in the end of the study and further in vivo studies are warranted to corroborate the results of this study.

Reviewer #2: In this work, the authors evaluate the effects on the load of Huntington’s disease-related protein aggregates of semi-purified extracts obtained from fruits of 8 different genotypes of Chilean-native Ugni molinae berries and comparatively determine the possible differences in their phenolic composition.

They identified significant differences in the biomedical properties and polyphenolic composition of ACEs and ETEs obtained from U. molinae fruits of different genotypes. Among those extracts, ETE 19-1 treatment had a potent activity over the abnormal protein aggregation on two different cellular models of HD, which might be mediated, in part, by autophagy induction.

The authors in this work should verify the effect of the treatment with polyphenol extracts on some biological parameters that could be modified. The effects on cell proliferation and cell cycle should be evaluated (PMID: 31412320). Furthermore, considering the antioxidant properties of the extracted polyphenols, it is useful to check whether the levels of endogenous reactive oxygen species (ROS) undergo variations (PMID: 19212014).

The most abundant polyphenol found in the extracts analysed by the authors is the flavonoid Quercetin. Quercetin has been reported to act as a senolytic by selectively removing senescent cells (PMID: 29315311). Senescence is a process that occurs following genotoxic stimuli and induces permanent cell cycle arrest with a loss of cellular functions (PMID: 27288264). Recently, Quercetin has been shown to display senolytic effects in some primary senescent cells, likely as a consequence of its inhibitory effects on specific anti-apoptotic genes, displaying senescence delaying activity in primary cells and rejuvenating effects in senescent cells (PMID: 26343116: PMID: 33242601; PMID: 32686219). Based on these knowledges, it is of fundamental importance to evaluate the effect of polyphenol extracts on cellular senescence in in vitro studies by beta-galactosidase assay and by western blot analysis of the levels of expression of various proteins involved in senescence and in cell cycle exit such as RB - RB2 - p107 - p53 - p27 -p21 - ARF - p16. (PMID: 26498687). These experiments are crucial to understand if with this treatment there is a reduction in cellular senescence that can have a role in contrasting the accumulation of misfolded proteins in the brain.

6. PLOS authors have the option to publish the peer review history of their article (what does this mean?). If published, this will include your full peer review and any attached files.

Reviewer #1: No

Reviewer #2: No

---

## [Author Response · Author response to Decision Letter 0]

9 Jun 2021

Branislav T. Šiler, Ph.D. 

Academic Editor PLOS ONE

Re: PONE-D-21-01842

Dear Dr. Šiler,

Thank you very much for conveying the reviewers’ thoughtful and constructive comments on our manuscript entitled “A phenolic-rich extract from Ugni molinae berries reduces abnormal protein aggregation in a cellular model of Huntington’s disease” submitted for publication as a research article in PLOS ONE. 

We appreciate the overall enthusiasm and interest in response to our work. We believe that reviewers appear to be supportive of the study and make precise suggestions to improve the quality of the manuscript. We prepared a point-by-point response to their comments along with a revised version of the manuscript, addressing virtually all reviewers’ suggestions to comply with PLOS ONE’s publication criteria. We believe our manuscript has been strengthened to highlight the novelty and relevance of the work.

The current study demonstrates the anti-aggregative activity of a semipurified extract from a Chilean native berry called U. molinae on Huntington’s Disease (HD) cellular models. Using spectrophotometric, chromatographic, and cell-based assays, we comparatively explored the chemical, biological and pharmacological properties of 16 different extracts obtained from fruits of diverse U. molinae genotypes. The citotoxicity and anti-aggregative activity of these extracts were assesed through an automated fluorescent microscopy platform, which allowed the rapid and unbiased analysis of the samples. This approach resulted in the identification of 5 genotypes among the ethanolic extracts which showed anti-aggregative activity without citotoxicity. Furthermore, we observed that the polyphenolic profile among these murtilla ethanolic extracts has similarities in concentration and occurrence but, through further validation, we determined only one genotype (19-1) whose polyphenolic-rich ethanolic extract showed a promising and unanticipated effect over abnormal protein aggregation related to neurodegenerative diseases using 2 different cellular models of HD.

We would like to apologize for the delay in the resubmission of our manuscript but, although we are not facing a full quarantine in Santiago (Chile) at the moment, sanitary measures due to the current pandemic only allow research labs to be partially operative. Nonetheless, we were able to address the reviewers’ concerns regarding our manuscript. We analysed the effects of the 19-1 ethanolic extract on cell proliferation, cell cycle, reactive oxygen species and cellular senescence, thus, we hope that this version of the manuscript will receive a positive feedback. We would also like to acknowledge your time and effort in handling this manuscript, and we hope that you will find that the revised manuscript deserves publication in PLOS ONE.

Response to reviewers:

Reviewer #1: 

The present study has studied the effect of phenolic rich extract of Ugni molinae berries against protein aggregation in neurodegenerative disorders especially HD. The study has been very well designed and all the methods used in this study have been explained in sufficient details. The results and discussion of the present manuscript is also very well articulated and explicitly explained. Following are some of my observations which require some clarity:

Comment 1. Authors have used ‘simple maceration’ for extracting the plant material. What was the duration of extraction and also mention the solvent to material ratio as these are important parameters that will affect the extraction efficiency. Since the effect of extracts is correlated with the phenolic content, these parameters are important for reproducibility of the results.

Answer: We would like to thank the reviewer for the possitive and constructuve comments and for highlighting of the relevance of our work.

 Regarding the extraction of the plant material, we performed a sequential extraction with solvents of increasing polarity (solvent-to-material ratio was 1:5) until complete exhaustion of the plant material, according to protocols previously validated in our lab (PMID: 27542470). Complete exhaustion means that the plant material was macerated until we observed a loss in the pigmentation of the extraction solvent. We also performed a cualitative analysis by thin layer chromatography on silica gel GF254 (Merck 5554) and checked for any positive spots under UV light and after spraying with different reagents as Liebermann–Burchard, anisaldehyde and NP/PEG reagent. If no signals were detected, then the extraction process continued with the next solvent. As requested, these parameters are now included in the Materials and Methods section of the manuscript’s revised version (Page 9, line 155). 

Comment 2. Why 1% formic acid in ethanol was used for extraction?

Answer: We thank the reviewer for pointing this issue. As we mentioned before, for the extraction of plant material, we used solvents of increasing polarity. We performed the extraction sequentially in order to obtain the dichloromethane extracts (DCMEs), acetone extracts (ACEs), and ethanolic extracts (ETEs). Ethanol was mixed with a low percentage of formic acid in order to obtain the ETEs. Other studies have observed the presence of anthocyanins in the murtilla fruit (PMID 25838172; 33981771; 31935880). Anthocyanins can be extracted from plant material with polar solvents, as acetone and ethanol. As anthocyanins are more stable in acidic media, we decided to include a mixture formic acid (1%) and ethanol in our protocol to extract any anthocyanin that would not be extracted in acetone. These details are mentioned in the Materials and Methods section of the manuscript’s revised version (Page 9, line 152).

Comment 3. Authors have stated that polyQ79-EGFP inclusions are not correlated with the TPC of different ETEs but with polyphenolic content. How you differentiate between total phenolic and polyphenolic content? Further, ETE 19-2 exhibited highest phenolic content, what was its effect on polyQ79-EGFP inclusions?

Answer: The Total Phenolic Content (TPC), measured by the Folin-Ciocalteu method, is significantly different among murtilla genotypes (Table 1). Through liquid chromatography coupled to mass spectrometry we observed that, even if there are differences in the TPC (Table 1) and their relative concentration (Table S1), the phenolic compounds present in the extracts are similar: Flavonoid glycosides, ellagitannins, and phenolic acids (Figs 6, 7 and Table S1). Interestingly, we observed that only one genotype stood up having significant effects over both polyQ79 and mHTT aggregation: ETE 19-1. Thus, the differences observed on polyQ aggregation might be due to the specific mixture of compounds found in the murtilla extracts and not the concentration of polyphenols. 

Regarding ETE 19-2, in an unbiased screening using the Cellomics machine we observed that treatment with this extract did not reduce polyQ79 aggregation significantly (Fig 2) and was not selected for secondary validation. 

These issues are stated stated in our manuscript in Page 23, line 421; Page 23, line 429; Page 24, line 437; Page 26, line 470; Page 29, line 555; Page 36, line 676. 

Comment 4. The results of the extracts have been mainly correlated with the phenolic compound especially, flavonoid glycosides. But the effects of different extracts are not in concordance with the TPC. Authors have mentioned that, the effects of ETE 19-1 are attributed to a specific combination of phenolic compounds present in the extract. Was the ETE free from other type of phytoconstituents? Since constituents belonging to other phytochemical class are also soluble in ethanol, there may be possibility that phytoconstituents belonging to other classes might play a role in the activity. If yes then this should also be mentioned and not only the phenolic compounds.

Answer: We thank the reviewer for pointing out this ideas. Through the chemical profile analysis performed in our study (Table 1, Figs 6 and 7, and Table S1), it is not possible to confirm the absence of other types of phytoconsitutents in the plant extracts, including the ETE 19-1. But, as metioned earlier, we performed a sequential extraction with solvents of increasing polarity in order to obtain the extracts with dichloromethane (DCMEs), acetone (ACEs), and ethanol (ETEs). This process allowed us to concentrate the polyphenolic compounds in the ETEs reducing the influence of other type of phytoconstitutens in their pharmacological properties. As requested, we now include additional information in the revised version of the manuscript (Page 30, line 565)

Comment 5. The only limitation of the study is the correlation of the activity of the berries seed extract with polyphenolic compounds only on basis in-vitro studies only. This should be mentioned in the end of the study and further in vivo studies are warranted to corroborate the results of this study.

Answer: We thank the reviewer for this comment. We have included this modification in the new version of our manuscript (Page 35, line 663). 

We deeply appreciate the comments from this reviewer to better explain our findings and improve the description of our results.

Reviewer #2: 

In this work, the authors evaluate the effects on the load of Huntington’s disease-related protein aggregates of semi-purified extracts obtained from fruits of 8 different genotypes of Chilean-native Ugni molinae berries and comparatively determine the possible differences in their phenolic composition. They identified significant differences in the biomedical properties and polyphenolic composition of ACEs and ETEs obtained from U. molinae fruits of different genotypes. Among those extracts, ETE 19-1 treatment had a potent activity over the abnormal protein aggregation on two different cellular models of HD, which might be mediated, in part, by autophagy induction.

Comment 1. The authors in this work should verify the effect of the treatment with polyphenol extracts on some biological parameters that could be modified. The effects on cell proliferation and cell cycle should be evaluated (PMID: 31412320). Furthermore, considering the antioxidant properties of the extracted polyphenols, it is useful to check whether the levels of endogenous reactive oxygen species (ROS) undergo variations (PMID: 19212014). 

Answer: We would like to thank the reviewer for his/her positive comments of our work and ideas to improve the quality study. We considered these comments seriously and performed a series of new assays that involved the participation of additional investigators in the lab with experience in cell signaling and cell biology. Thus, we evaluated the effects of ETE 19-1 treatment over cell proliferation, cell cycle, and levels of endogenous reactive oxygen species (ROS) (Fig S2). The results indicate that ETE 19-1 tested at two different concentrations did not affect cell proliferation in time course experiments by counting cells (Fig. S2B). Similar results were obtained using the MTS assay (Fig. S2C). Quercetin was also tested as control. Cell cycle was analyzed using P staining in permeabilized cells followed by FACS analysis. Again, ETE 19-1 did not affect cell cycle, whereas quercetin had small effects (Fig. S2D). or ROS production. Finally, we evaluated ROS content in cells treated with ETE 19-1 or quercetin using DHE staining followed by FACS analysis. No differences in ROS levels were detected. As positive control cells were treated with H2O2, observing that quercetin slightly reduced ROS levels (Fig. S2E). 

We also evaluated if ETE 19-1 treatment activates the DNA damage response (DDR) (Fig S3). We measured two canonical markers of the pathway by assessing the levels of checkpoint kinase 1 (CHK1) phosphorylation (Fig. S3A). As positive control, cells were treated with the DNA damaging agent etoposide. In addition, we performed real time RT-PCR to measure the mRNA levels of p21, a canonical target of p53. Again, we did not observe induction of p21 in cells treated with ETE 19-1 (Fig. S32). A higher percentage of HEK293 cells treated with quercetin were arrested in G0/G1 phase compared to control and ETE 19-1, which means that treatment with this polyphenol might be inhibiting normal cell growth (Fig S2D). Supporting this, quercetin treatment also induced a higher expression of p21 (Fig S3B), meanwhile ETE 19-1 treatment did not induce a higher expression of p21 and pCHK1 (Figs S3A and S3B).

To corroborate the chemical integrity of the ETE 19-1 extract for these new experiments, we also evaluated its total phenolic content (TPC) and we observed that there have not been changes in the TPC of the extract since the first evaluation was performed (Fig S2A). Overall, these results confirm that ETE 19-1 treatment is not cytotoxic and that its effects over protein aggregation are not related to the modulation of cell proliferation or DNA damage. Furthermore, as quercetin induced arrest in G0/G1 phase and a higher expression of p21, ETE 19-1 represents a safer option for the reduction of protein aggregation. 

Comment 2: The most abundant polyphenol found in the extracts analyzed by the authors is the flavonoid Quercetin. Quercetin has been reported to act as a senolytic by selectively removing senescent cells (PMID: 29315311). Senescence is a process that occurs following genotoxic stimuli and induces permanent cell cycle arrest with a loss of cellular functions (PMID: 27288264). Recently, Quercetin has been shown to display senolytic effects in some primary senescent cells, likely as a consequence of its inhibitory effects on specific anti-apoptotic genes, displaying senescence delaying activity in primary cells and rejuvenating effects in senescent cells (PMID: 26343116; PMID: 33242601; PMID: 32686219). Based on these knowledges, it is of fundamental importance to evaluate the effect of polyphenol extracts on cellular senescence in in vitro studies by beta-galactosidase assay and by western blot analysis of the levels of expression of various proteins involved in senescence and in cell cycle exit such as RB - RB2 - p107 - p53 - p27 -p21 - ARF - p16. (PMID: 26498687). These experiments are crucial to understand if with this treatment there is a reduction in cellular senescence that can have a role in contrasting the accumulation of misfolded proteins in the brain.

Answer: We would like to thank the reviewer for highlighting this point. As requested, we performed additional experiments to assess the effects of ETE 19-1 treatment on cellullar senescence. Senescent cells were induced and then treated with 3 different concentrations of ETE 19-1 (50, 100 and 200 ug/mL) for 24 and 48 hr. Given that ETE 19-1 has a complex mixture of polyphenols (Figures 6, 7, and Table S1) and they have been described as inductors of senolysis through apoptosis (PMID: 32997601), we evaluated the presence of the apoptotic marker cleaved caspase 3 (CST cod. 9669S) by immunofluorescence. As shown in Figure S4, no significant difference between the control (time 0), and exposed cells (times 24, and 48 h) were found. Thus, ETE 19-1 treatment does not induce a reduction in cellular senescence that could be explaining its effects on the accumulation of misfolded proteins.

We would like to thank again this reviewer for the ideas to improve the cell biology characterization of our study.

---

## [Editor Report · Decision Letter 1]

16 Jun 2021

PONE-D-21-01842R1

A phenolic-rich extract from Ugni molinae berries reduces abnormal protein aggregation in a cellular model of Huntington’s disease

PLOS ONE

Dear Dr. Delporte,

Thank you for submitting your manuscript to PLOS ONE. After careful consideration, we feel that it has merit but does not fully meet PLOS ONE’s publication criteria as it currently stands. Therefore, we invite you to submit a revised version of the manuscript that addresses the points raised during the review process.

The authors have properly addressed the reviewers concerns. However, since PLOS ONE does not copyedit manuscripts, authors' action is needed in order to improve typographic and grammar errors. Please, double-check the whole text in this regard. Some hints:

L130 and elsewhere: Please use 1×, not 1X.

L131 and further: CO2, O2, CaCl2 - "2" should stand in subscript.

L199, L216: Please do not start a sentence with a number. Rephrasing is needed. Use "×" instead of "x".

L249 and elswhere: Please avoid italicizing expressions which do not need it. For instance, "ReSpect" is stated in both Roman and cursive.

Figure 6: "Pentoside" and "Hexoside" should not be capitalized.

Main title: "from" should not stand italicized.

L36: Please introduce the common name "murtilla" since used further in the manuscript.

L38 and further: Once introduced in full, species' Latin name should be abbreviated as *U. molinae*.

We look forward to receiving your revised manuscript.

Kind regards,

Branislav T. Šiler, Ph.D.

Academic Editor

PLOS ONE
---

## [Author Response · Author response to Decision Letter 1]

1 Jul 2021

Dear Lilla Petho, 

Thank you very much for raising this issue with our revised manuscript. As instructed, in this letter we include our amended financial statements.

We would like to add the following funding sources to our Funding Statement:

FONDECYT 1180186 (CH)

FONDECYT 3190738 (AS)

FONDECYT 3210294 (PP)

FONDECYT 1200255 (CC). 

In addition, we would also like to thank the support from the U.S. Air Force Office of Scientific Research FA9550-16-1-0384, US Office of Naval Research-Global (ONR-G) N62909-16-1-2003 (CH).

Thus, our Funding Statement should read as follows:

“This work was financially supported by ANID-Chile (Agencia Nacional de Investigación y Desarrollo - www.ANID.cl) grants for Doctoral studies N° 21150769 (RPA) and N°21150851 (JLO), FONDECYT 1130155 (CDV), FUNDACION COPEC-UC grant 2013.R.40 (AR), FONDEQUIP EQM120164 (CH), and Millennium Institute P09-015-F (RLV/CH). We also thank the support from ANID/FONDAP program 15150012 (RLV/CH), FONDECYT 1191003 (RLV), FONDECYT 1180186 (CH), FONDECYT 3190738 (AS), FONDECYT 3210294 (PP), FONDECYT 1200255 (CC), FONDECYT 1140549, FONDEF ID16I10223, FONDEF ID11E1007, CONICYT-Brazil 441921/2016-7, Michael J Fox Foundation for Parkinson´s Research – Target Validation grant 9277 (www.michaeljfox.org/) (CH), European Commission R&D MSCA-RISE 734749 (ec.europa.eu) (CH). In addition, we would like to thank the support from the U.S. Air Force Office of Scientific Research FA9550-16-1-0384, US Office of Naval Research-Global (ONR-G) N62909-16-1-2003 (CH).”

Finally, we would also like to acknowledge your time and effort in handling this manuscript. We hope that this revised version complies with PLOS ONE’s submissions guidelines. Please, do not hesitate to contact us if additional information or changes are required.

Best regards,

Carla Delporte, Ph.D.

Corresponding Author

Claudio Hetz, Ph.D.

Corresponding Author

---

## [Editor Report · Decision Letter 2]

5 Jul 2021

A phenolic-rich extract from Ugni molinae berries reduces abnormal protein aggregation in a cellular model of Huntington’s disease

PONE-D-21-01842R2

Dear Dr. Delporte,

We’re pleased to inform you that your manuscript has been judged scientifically suitable for publication and will be formally accepted for publication once it meets all outstanding technical requirements.

Kind regards,

Branislav T. Šiler, Ph.D.

Academic Editor

PLOS ONE
---

## [Editor Report · Acceptance letter]

21 Jul 2021

PONE-D-21-01842R2 

A phenolic-rich extract from *Ugni molinae* berries reduces abnormal protein aggregation in a cellular model of Huntington’s disease 

Dear Dr. Delporte:

I'm pleased to inform you that your manuscript has been deemed suitable for publication in PLOS ONE. Congratulations! Your manuscript is now with our production department. 

Kind regards, 

on behalf of

Dr. Branislav T. Šiler 

Academic Editor

PLOS ONE